# Potential for Core Fucose-Targeted Therapy Against HBV Infection of Human Normal Hepatocytes

**DOI:** 10.3390/v17091242

**Published:** 2025-09-15

**Authors:** Shinji Takamatsu, Chiharu Morita, Daisuke Sakon, Kotaro Nakamura, Honoka Hishii, Jumpei Kondo, Keiji Ueda, Eiji Miyoshi

**Affiliations:** 1Department of Molecular Biochemistry and Clinical Investigation, Graduate School of Medicine, The University of Osaka, 1-7 Yamada-oka, Suita 565-0871, Osaka, Japanjumpeko@sahs.med.osaka-u.ac.jp (J.K.); 2Department of Virology, Graduate School of Medicine, The University of Osaka, 2-2 Yamada-oka, Suita 565-0871, Osaka, Japan; cmorita@virus.med.osaka-u.ac.jp (C.M.); kueda@virus.med.osaka-u.ac.jp (K.U.); 3Department of Microbiology, Juntendo University School of Medicine, 2-1-1 Hongo, Bunkyo-ku, Tokyo 113-8421, Japan

**Keywords:** HBV, PhoSL, glycotherapy, Fut8, PXB cells

## Abstract

Core fucose is one of the most important glycans in HBV infection. In this study, we investigated whether *Pholiota squarrosa* lectin (PhoSL), a lectin that specifically binds to core fucose, exerts an inhibitory effect in an HBV infection model of normal human hepatocytes. Similarly to previous studies using hepatocellular carcinoma cells (HepG2-C4), the coexistence of PhoSL during HBV infection inhibited HBe antigen production and HBV cccDNA in normal human hepatocytes in a PhoSL concentration-dependent manner. Furthermore, this effect of PhoSL was found to be able to suppress HBe antigen production in a treatment period-dependent manner, even when PhoSL was administered after HBV infection. Our previous research has revealed that the mechanism by which PhoSL inhibits HBV infection is through physical inhibition by binding to the HBV receptor and inhibition of HBV entry into cells by inhibiting the phosphorylation of EGFR, a co-receptor for NTCP. Furthermore, this study suggested that PhoSL may also inhibit HBV proliferation in cells through other mechanisms that require further investigation. PhoSL is a lectin, derived from edible *Pholiota squarrosa* (shaggy scalycap) mushrooms, that is resistant to acid and heat. In addition, it has a low molecular weight and can be chemically synthesized, so it is expected to be used clinically as a new carbohydrate therapy for HBV in the future.

## 1. Introduction

Hepatitis B virus (HBV) is a major hepatitis virus. This infection is a serious global health problem, with 2 billion people infected worldwide and 350 million suffering from chronic HBV infection, although the incidence rate varies greatly between countries [1]. The main route of infection is mother-to-child transmission of HBV, which is established through the bloodstream. As chronic hepatitis caused by HBV progresses, the incidence of liver cirrhosis and hepatocellular carcinoma increases, so aggressive treatment of HBV is recommended [2,3]. Currently, interferon and nucleoside analogues are commonly used for treatment, but the former has a low response rate and side effects, and the latter is problematic because the drug cannot be discontinued [4,5]. Therefore, there is a need for the development of drugs such as direct-acting hepatitis C antivirals (DAAs) that are highly effective in treating hepatitis B virus and do not cause permanent recurrence of the virus even if treatment is discontinued [6,7,8]. In 2012, sodium taurocholate cotransporting polypeptide (NTCP, encoded by SLC10A1) was identified as a functional receptor for HBV by a Chinese group [9]. NTCP is a receptor involved in the enterohepatic circulation of bile acids [10]. To date, several compounds have been reported as HBV entry inhibitors targeting NTCP and are undergoing clinical trials [11,12,13]. Because the HBV receptor is present on the cell surface, possibly as a complex like the hepatitis C virus receptor [14,15,16], such coreceptors may be new targets for HBV treatment.

Many cellular proteins, especially secreted and cell surface proteins, are glycosylated and are known to be involved in the biological functions of cells [17,18,19]. Since many cell surface proteins, including viral receptors and viral envelope proteins, are glycoproteins, glycans have become targets for antiviral therapy [20,21]. Changes in the glycosylation state of proteins may affect the infectivity of virus particles [22]. For example, influenza infection requires the cleavage of sialic acid from certain proteins by sialidases on host cells [23]. The inhibition of sialidases has been a target for anti-influenza therapy [20]. The sialylation of host cells is also required for SARS-CoV-2 infection [24]. Based on these findings, many lectins, i.e., proteins that selectively bind to specific glycans, are considered to be potential candidates for antiviral drugs [25]. For example, the lectin Griffithsin can bind to mannose-rich oligosaccharides and inhibit human immunodeficiency virus (HIV)-1 infection without cytotoxicity [26]. It is also known that human surfactant protein A can inhibit SARS-CoV-2 infection [27].

NTCP is a glycoprotein with four N-linked glycosylation sites [28]. We previously revealed that core fucose is one of the glycosylation modifications most involved in HBV infection using HBV pseudoparticle BNCs (bionanocapsules) [29]. Core fucose is fucose added to the innermost N-acetylglucosamine of N-glycans and specifically synthesized by α1-6 fucosyltransferase Fut8 [30]. Furthermore, we recently reported that PhoSL (*Pholiota squarrosa* lectin), which specifically recognizes core fucose, inhibits HBV infection and cccDNA synthesis in a concentration-dependent manner in HepG2 cells overexpressing NTCP (HepG2-C4) [31]. Imaging experiments demonstrated that the mechanism by which PhoSL inhibits HBV infection occurs not only by directly binding to NTCP but also by inhibiting EGF receptor phosphorylation. Furthermore, the inhibition of HBV infection by PhoSL was observed even in Fut8 knockout C4 cells, suggesting that PhoSL also binds to HBV itself, is taken up into the cell, and has some effect on the growth and replication of HBV.

PhoSL has been purified from edible mushrooms and has been confirmed to be non-toxic when administered intraperitoneally and orally to animals [32]. In this study, we investigated whether PhoSL also exerts an inhibitory effect on HBV infection in normal human hepatocytes, aiming at future clinical drug discovery. As a result, we found that when PhoSL was administered simultaneously with HBV infection, it could suppress HBe antigen production and cccDNA synthesis in normal human hepatocytes in a concentration-dependent manner. Furthermore, even when PhoSL was administered after HBV infection was established, the inhibitory effect of HBV infection by PhoSL was found to be able to suppress HBe antigen production and cccDNA synthesis in a treatment-period-dependent manner.

## 2. Materials and Methods

### 2.1. Cells

PXB cells were purchased from PhoenixBio (Hiroshima, Japan). These are fresh hepatocytes harvested from human hepatocyte chimeric mice by collagenase perfusion. The purity of human hepatocytes in PXB cells is approximately 93%, and the donor cells are derived from a 1-year-old Caucasian boy or a 12-year-old Caucasian girl.

PXB cells were cultured in DMEM (nacalai-tesque, Kyoto, Japan) containing the following: 10% fetal bovine serum (FBS) (Nichirei, Tokyo, Japan) with 20 mM HEPES (Thermo Fisher, Waltham, MA, USA); 44 mM NaHCO_3_ (Fujifilm-Wako, Osaka, Japan); 100 U/mL Penicillin G (Thermo Fisher); 100 µg/mL Streptomycin (Thermo Fisher); 15 µg/mL L-proline (Fijifilm-Wako, Osaka, Japan); 0.25 µg/mL Insulin (Sigma-Aldrich, St Louis, MO, USA); 50 nM Dexamethasone (Sigma-Aldrich); 5 ng/mL EGF (Sigma-Aldrich); 0.1 mM L-ascorbic acid 2-phosphate (Fujifilm-Wako); and 2% DMSO (Sigma-Aldrich). This was carried out in a 37 °C incubator under a humidified atmosphere containing 5% CO_2_. HepG2.2.15 cells were provided by Dr. T. Okamoto (Juntendo University School of Medicine) and cultured in DMEM supplemented with 10% fetal bovine serum (FBS), 100 U/mL penicillin, and 100 µg/mL streptomycin [33].

PhoSL was obtained from Dr. Y. Kobayashi (Mitsubishi Chemical, Tokyo, Japan) and Myristoylated PreS1 peptide (myrPreS1) was synthesized by Scrum (Tokyo, Japan) [34].

### 2.2. Preparation of HBV Particles

HepAD38.7 cells are a tetracycline-regulated HBV-producing cell line that use the Tet-Off system. To obtain HBV particles for HBV infection experiments, tetracycline was removed from the culture medium and HBV replication was induced in HepAD38.7 cells. The culture medium of confluent HepAD38.7 cells cultured without tetracycline was collected every week for 2 weeks, and HBV particles were precipitated by adding PEG 8000 (final concentration 6%) and allowing it to stand overnight at 4 °C. The precipitate was centrifuged, resuspended in phosphate-buffered saline (PBS), concentrated, and filtered through a 0.45 µm filter (Millipore, Burlington, MA, USA). HBV DNA was quantified by qPCR and adjusted to 10^7^ HBV/µL.

### 2.3. Cytotoxicity Assay

PXB cells seeded at 4 × 10^5^ cells/well in 24-well plates were cultured at 37 °C under humidified conditions with 5% CO_2_. After replacing with fresh medium, cells were co-cultured with HBV with PhoSL (0, 1, 2.5, 5, or 10 μg/mL) for 1 day and then continued to be cultured for an additional 3, 6, 9, or 12 days, with medium changes every 3 days. At each time point, culture supernatants were harvested, diluted 25-fold in LDH storage buffer, and stored at −20 °C until analysis. Viability was measured using the LDH-Glo Cytotoxicity Assay (Promega, Madison, WI, USA) according to the manufacturer’s instructions. Chemiluminescence intensity was measured using the GloMax Discover System (Promega).

### 2.4. HBV Infection and HBe Antigen Measurement

HBV stocks were prepared as previously described [35]. PXB cells (4 × 10^5^/well) seeded in 24-well plates were replaced with 0.5 mL of fresh medium once upon arrival, and HBV infection experiments were started the next day or later. The cells were infected with HBV (2 × 10^7^ genomes/well [approximately 50 genome infectious equivalents (G.E.I)]) in a medium containing 4% polyethylene glycol 8000 (PEG 8000 [Sigma-Aldrich]). After 24 h of incubation, the cells were washed twice with 0.5 mL of PBS and cultured for 12 days in 0.5 mL of the same medium, with the medium changed every 3 days. Culture supernatants were collected five times, on the day of medium replacement and at the end of incubation, and hepatitis B e antigen (HBeAg) was quantified by using an HBeAg ELISA kit (Bioneovan, Beijing, China). For a positive control, pattern 1 was used, in which myrPS1 was co-administered at a final concentration of 1 µM for 1 day at the time of HBV infection. PhoSL was administered at a final concentration of 10 µg/mL in the following four patterns: (1) administration for 12 days after a medium change following the end of HBV infection; (2) administration for the final 9 days of culture, (3) administration for the final 6 days of culture, and (4) administration for the final 3 days of culture.

For a negative control, medium replacement was performed after HBV infection, but no PhoSL was added. In an experiment to examine the concentration dependence of PhoSL, PXB cells were infected with HBV for one day, washed out with PBS, and then cultured for 12 days with medium containing 0, 1, 2.5, 5, or 10 µg/mL PhoSL, with the medium changed every three days.

### 2.5. Preparation of Extracellular HBV Particle-Associated DNA

Extracellular core particle-associated HBV DNA was prepared as follows: 30% PEG8000 (Sigma-Aldrich) was added to the collected culture supernatant to a final concentration of 6%, and the mixture was mixed by end-over-end at 4 °C overnight. After centrifugation at 8000× *g* for 30 min at 4 °C, the supernatant was removed, and 450 µL of TBS (20 mM Tris-HCl, pH 7.8, 150 mM NaCl) was added to the precipitate, which was then dissolved by end-over-end mixing at 4 °C overnight. After centrifugation at 8000× *g* for 30 min at 4 °C, the supernatant was collected and incubated overnight at 37 °C with 12.5 U DNase I (TaKaRa, Kusatsu, Japan) and 500 ng RNase (Roche, Basel, Switzerland) in 1 × DNase buffer (40 mM Tris-HCl, pH 7.9, 10 mM NaCl, 6 mM MgCl2, 10 mM CaCl2, 2 mM DTT) to degrade DNA outside the virions. EDTA (pH 8.0) was added to a final concentration of 10 mM, and the mixture was heated at 80 °C for 5 min to inactivate DNase I. In total, 50 µL of this reaction mixture was mixed with 350 µL of Proteinase K reaction buffer (10 mM Tris-HCl, pH 7.8, 10 mM EDTA, 0.5% SDS) and 4 µL of 600 mU/µL Proteinase K (nacalai-tesque), and digested overnight at 56 °C. An equal volume of phenol/chloroform/isoamylalcohol (25:24:1) was added, mixed using a vortex and centrifuged at 13,500× *g* for 30 min at 4 °C, and 300 µL of the aqueous layer was collected. Further, 2 µL of 10 mg/mL glycogen (nacalai-tesque), 30 µL of 3 M sodium acetate, pH 5.2 (Sigma-Aldrich), and 750 µL of ethanol (nacalai-tesque) were added, mixed using a vortex, and centrifuged at 13,500× *g* for 30 min at 4 °C, and the supernatant was removed. In total, 1000 µL of 70% ethanol was added, and the mixture was centrifuged at 13,500× *g* for 15 min at 4 °C, after which the supernatant was removed. The precipitate was air-dried briefly and then dissolved in 20 µL of TE.

### 2.6. Preparation of Hirt DNA and Quantification of HBV DNA Copies

Hirt DNA was isolated from HBV-infected PXB cells using a standard Hirt extraction method. Briefly, 350 µL of 1% SDS/TE was added to one well of cells and incubated for 10 min, and the lysate was scraped off with a pipette tip. Then, 50 µL of 0.5 M NaCl was then added to the lysate, which was then mixed and incubated overnight at 4 °C. After centrifugation at 18,000× *g* for 30 min at 4 °C, 200 µL of the supernatant was collected. To the supernatant, 1 µL of 500 µg/mL RNase (Roche, Indianapolis, IN, USA) was then added, and this was mixed and incubated at 65 °C for 30 min. An additional 2 µL of 0.2 mg/mL proteinase K (Roche) was added, mixed, and incubated at 56 °C overnight. After adding 200 µL of TE-saturated phenol (nacalai-tesque, Kyoto, Japan), the mixture was mixed by vortexing and centrifuged at 18,000× *g* for 20 min at 4 °C, and 200 µL of the supernatant was collected. Then, 200 µL of phenol-chloroform-isoamyl alcohol (PCI)25:24:1 (nacalai-tesque) was added to the supernatant, which was mixed by vortexing and centrifuged at 18,000× *g* for 30 min at 4 °C, and 150 µL of the supernatant was collected. The following were also added to the supernatant, followed by centrifugation at 20,000× *g* for 30 min at 4 °C: 2 µL of 10 mg/mL glycogen (Takara Bio, Shiga, Japan), 15 µL of 3 M CH_3_COONa, and 375 µL of ethanol. After removing the supernatant, 550 µL of 70% ethanol was added and centrifuged at 20,000× *g* for 15 min at 4 °C. The supernatant was removed, and the samples were briefly air-dried before dissolving in 20 µL of TE. cccDNA measurements were performed by digesting 10 µL of dissolved Hirt DNA with 0.5 µL of T5 exonuclease (Promega, Madison, WI, USA) for 60 min at 37 °C.

Extracellular and intracellular HBV DNA was quantified by real-time PCR. Quantification was performed by creating a standard curve using a dilution series of HBV DNA from 1.2 × 10^8^ to 1.2 × 10^3^ genome copies, and real-time qPCR was performed using a QuantStudio1 real-time PCR system (Thermo Scientific). The primers used were HBs-qPCR-Fw: 5′-CTTCATCCTGCTGCTATGCCT-3′ and HBs-qPCR-Rv: 5′-AAAGCCCAGGATGATGGGAT-3′, both at a concentration of 10 µM. After initial denaturation at 95 °C for 30 s, PCR reactions consisted of two steps: denaturation at 95 °C for 5 s and annealing and extension at 60 °C for 10 s, with 40 cycles being performed.

HBV cccDNA was quantified by digital PCR. The amount of RNase P gene was measured using the TaqMan RNase P control reagent kit (Thermo, Waltham, MA, USA) and used as an internal control. Digital PCR was performed using a QuantStudio Absolute Q Digital PCR System (Thermo Fisher, Waltham, MA, USA). The primers used were as follows: Probe (FAM) 5′-CTGTAGGCATAAATTGGT-3′,

cccDNA_Fwd: 5′-CGTCTGTGCCTTCTCATCTGC-3′, and cccDNA_Rev: 5′-GCACAGCTTGGAGGCTTGAA-3′.

An HBV production suppression experiment was performed using PhoSL in HepG2.2.15 cells.

Cells were seeded at 1 × 10^4^ cells per well in a 24-well plate. The next day, PhoSL was added at concentrations of 0, 1, 2.5, 5, and 10 µg/mL, along with a medium change. Three days later, the medium was changed and PhoSL was added in the same manner. After another three days of culture, the culture medium and cells were harvested. To assess the cytotoxicity of PhoSL, cells cultured for 3 and 6 days were treated with Trypsin-EDTA solution to detach them, suspended in medium, and mixed 1:1 with 0.4% Trypan Blue Stain (Thermo) in a 1:1 ratio. The cell count was measured using an automated cell counter (Countess 3 FL, Thermo Scientific), and the cell viability was estimated.

### 2.7. Statistical Analysis

Data are presented as the mean ± SD, and significance was tested using Student’s *t*-test with Microsoft Excel software. *p* < 0.05 was considered to indicate statistical significance.

## 3. Results and Discussion

### 3.1. PhoSL Inhibited HBV Infection in Normal Human Hepatocytes in a Concentration-Dependent Manner When Administered Simultaneously with HBV Infection

Figure 1A shows an outline of the PhoSL administration experiment.

First, to examine the concentration dependence of PhoSL, PhoSL was allowed to coexist during HBV infection. In our previous study, we found that PhoSL inhibited HBV infection in hepatocellular carcinoma HepG2-C4 cells. Therefore, to clarify whether PhoSL can also inhibit HBV infection in normal hepatocytes, we examined its effect using PXB cells. As shown in Figure 1B, PhoSL did not show significant toxicity to PXB cells up to a concentration of 10 µg/mL.

As shown in Figure 1C, PhoSL was able to suppress HBe antigen production in normal hepatocytes in a concentration-dependent manner, as well as in hepatocellular carcinoma cells. The results in Figure 1C show that the effect of PhoSL coexistence during HBV infection was large, and HBeAg production was dramatically suppressed during the measurement period from the third day after the end of infection. This is thought to be due to PhoSL’s strong inhibition of HBV intracellular entry.

We also attempted to measure intracellular cccDNA to investigate the concentration-dependent inhibitory effect of PhoSL on HBV replication, and although the results were for n = 1, we were able to confirm a significant inhibitory effect comparable to the results of HBe antigen measurements.

However, this is only the result when PhoSL coexists with HBV infection, and to predict the effect of PhoSL during natural HBV infection, it is necessary to take PhoSL continuously as a permanent preventive measure, which is not very realistic. If the infectious disease treatment effect of PhoSL can be confirmed after HBV infection, clinical application is expected. Therefore, using the same normal hepatic PXB cells, we investigated whether a therapeutic effect could be obtained by administering PhoSL for various periods starting one day after HBV infection.

### 3.2. PhoSL Suppressed HBeAg, Intra- and Extracellular HBV DNA, and cccDNA Production in a Treatment-Period-Dependent Manner, Even When Administered After HBV Infection

Figure 2A outlines an experiment in which PhoSL was administered therapeutically after HBV infection. Cells were infected with HBV for 1 day, washed twice with PBS, and then cultured in the presence of PhoSL for 3, 6, 9, or 12 days. Figure 2B shows the time course of HBe Ag produced in the culture supernatant after HBV infection. When myristoylated PreS1 peptide (myrPreS1) coexisted during HBV infection, the infection was strongly suppressed, and HBe Ag production in the medium was low even after 12 days.

To verify the feasibility of PhoSL therapy for patients with HBV infection, we used PXB cells to examine whether the administration of PhoSL after HBV infection has an infection-suppressing effect, using changes in HBe Ag production and extracellular HBV production as indicators. As shown in the results in Figure 2C, it was found that PhoSL can suppress HBe Ag production in a treatment time-dependent manner, even when administered after HBV infection. This result is almost reflected in the results in Figure 2D, and it was confirmed that the amount of extracellular HBV DNA was also suppressed with increasing PhoSL treatment time.

We found that PhoSL, when administered simultaneously with HBV infection, reduced the efficiency of HBV infection into PXB cells. Based on the results of previous studies, this is thought to be due to inhibition of entry, but in this study, we found that, even when treated after HBV infection, the amount of extracellular HBeAg and HBV DNA was suppressed in a treatment-time-dependent manner.

Previous studies have demonstrated that, even in the absence of HBV, PhoSL is internalized into cells, likely through binding to the core fucose of cell surface receptors [31].

To determine whether this was due to the inhibition of extracellular secretion or intracellular replication, we measured the amount of intracellular HBV DNA and cccDNA. The results are shown in Figure 2E,F. The measured intracellular HBV DNA includes HBV DNA in the cytoplasm, relaxed circular DNA (rcDNA) in the nucleus, and covalently closed circular DNA (cccDNA). If the amount of intracellular HBV DNA is unchanged with or without PhoSL administration, and only cccDNA is reduced, it would suggest that PhoSL affects the conversion of rcDNA to cccDNA. The conversion of rcDNA to cccDNA is an important step in the HBV life cycle, and cccDNA is the main template for HBV gene transcription, so it would be groundbreaking if PhoSL could inhibit cccDNA synthesis. However, since PhoSL treatment also reduced the total amount of HBV DNA in the cells, it is possible that the process of rcDNA translocation to the nucleus, denucleation, and the reimport of nucleocapsids into the nucleus is inhibited, but no conclusion has been reached at this time. While our current findings suggest that PhoSL inhibits the early steps of HBV infection, we were unable to determine whether this effect occurs at the level of enucleation, due to the limitations of our system. The cytoplasmic fraction contains both incoming and newly synthesized viral particles, and the amount of nuclear DNA was insufficient for Southern blot analysis. Future studies using replication-deficient systems such as HBV-NL [36] may help clarify the mechanism by distinguishing uncoating from other steps. Furthermore, core fucosylation may play an important role not only in protein import via endosomes, but also in secretion via the ER-Golgi system, and PhoSL may act on this system to reduce the amount of HBV released outside the cells. However, in a study using HepG2.2.15 cells established by introducing tandem repeats of the HBV genome into HepG2 cells, the 10 μg/mL concentration of PhoSL was slightly toxic after 6 days of incubation in HepG2.2.15 cells, as shown in Figure 3B. This was reflected in the results of reducing intracellular and extracellular HBV DNA levels, so the 10 μg/mL PhoSL-treated samples were excluded from the cccDNA assay. Considering the cytotoxicity results, it seems likely that PhoSL administration did not reduce the amounts of extracellular HBeAg, intracellular HBV DNA, extracellular HBV DNA, or intracellular cccDNA (Figure 3C–F). This cell line is a model for evaluating the release process of formed virus particles outside the cells at the late stage of HBV infection, as they can produce and secrete HBV particles inside the cells. It has been suggested that PhoSL does not affect the final release of HBV particles outside the cells. These results were also determined by using HB611 cells [37]. Even in HB611 cells, the inhibitory effect of PhoSL on HBV production was not confirmed. HB611 and HepG2.2.15 are similar in that they stably express HBV genes, but the difference is that HB611 is a cell line transfected with genotype C HBV DNA, while HepG2.2.15 is a cell line transfected with genotype D HBV DNA.

### 3.3. MyrPreS1 Peptide Did Not Affect the Amount of Extracellular HBeAg or the Amount of Intracellular and Extracellular HBV DNA and cccDNA Produced When Administered After HBV Infection

Another possible pathway for the suppressive effect of PhoSL on extracellular HBeAg and HBV DNA levels observed after HBV infection is the suppression of reinfection. Therefore, we investigated the effect of myrPreS1, which is known to be able to suppress the reinfection pathway. Figure 4A shows an outline of an experiment in which myrPreS1 was administered after HBV infection. Cells were infected with HBV for 1 day, washed twice with PBS, and then cultured for 12 days in the presence or absence of myrPreS1.

As shown in Figure 4B, myrPreS1 can strongly suppress HBV infection when it is coexisted during HBV infection, but when it is treated after infection, it cannot suppress the amount of extracellular HBe antigen, as shown in Figure 4C, and as shown in Figure 4D–F, no suppressive effect was observed on the amount of intracellular and extracellular HBV DNA and cccDNA, unlike in the case of PhoSL. In addition, PhoSL recognizes and binds to a glycan structure called core fucose, but since HBV glycosylation is affected by the balance of glycosyltransferases in the host cell, it is possible that core fucosylation is less likely to occur in PXB cells, which are normal hepatocytes. It has also been reported that Fut8 expression is higher in cancer cells than in normal cells [38,39]. Considering these findings together, it is speculated that the infection-suppressing effect observed when PhoSL was administered after HBV infection is not due to inhibition of reinfection, but rather due to PhoSL that was taken up into the cells by an as-yet-undetermined route and inhibited a step within the HBV replication pathway.

A brief summary of the results of this study is shown in Graphical Abstract. PhoSL binds to the core fucose of NTCP receptor and HBV particles, inhibiting HBV entry into cells by steric hindrance. PhoSL taken up into cells by endocytosis together with HBV may also inhibit cccDNA synthesis by some mechanism. However, PhoSL does not appear to affect the secretory pathway of HBV particles after HBV mRNA synthesis at the late stage of infection.

In this study, we demonstrated that PhoSL also has an inhibitory effect on HBV infection in normal hepatocytes. This inhibitory effect can suppress intracellular cccDNA levels in a time-dependent manner even when administered after HBV infection, and is therefore expected to have clinical applications in the future.

## Figures and Tables

**Figure 1 viruses-17-01242-f001:**
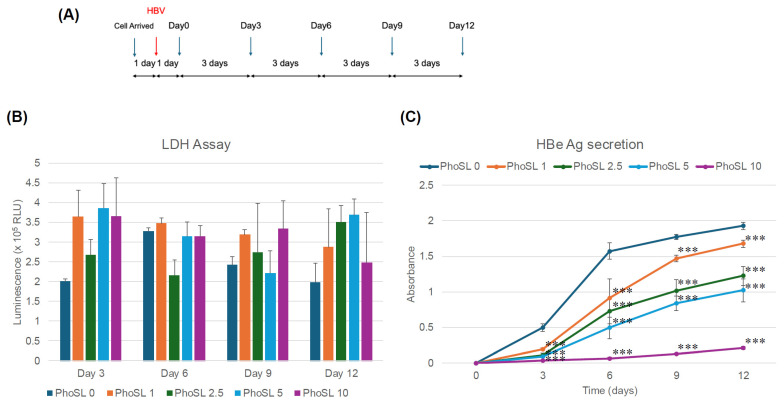
PhoSL inhibits the infection and replication of HBV in normal hepatocytes (PXB cells). (**A**) This figure shows an overview of the concentration-dependent experiments of PhoSL cotreatment during HBV infection. The blue arrow indicates the day of medium replacement, and the red arrow indicates the day of HBV infection. PhoSL was co-administered at various concentrations at the time of infection. (**B**) Cell viability of PXB cells cultured with PhoSL for 3, 6, 9, or 12 days is shown. PhoSL concentrations of 0, 1, 2.5, 5, and 10 μg/mL are shown in dark blue, orange, dark green, light blue, and magenta, respectively. Viable cells were measured by LDH-Glo cytotoxicity assay. No statistically significant differences were observed between 1, 2.5, 5, and 10 μg/mL PhoSL and 0 μg/mL PhoSL. Error bars indicate standard deviation; n = 3. (**C**) Time course of HBeAg production after treatment with various concentrations of PhoSL during HBV infection. Each data point is shown with the mean value +/− S.D. *** *p* < 0.001.

**Figure 2 viruses-17-01242-f002:**
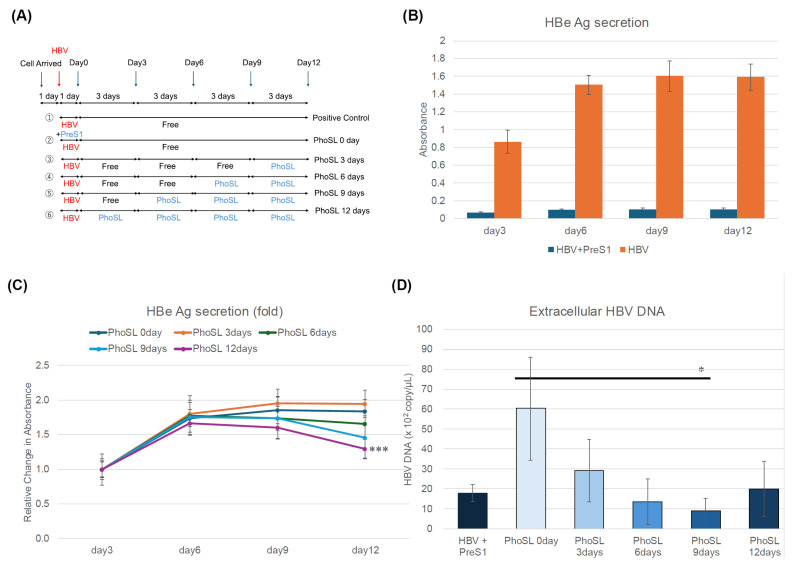
PhoSL suppressed extracellular HBeAg and HBV DNA production, as well as intracellular HBV DNA and cccDNA production in PXB cells in a treatment duration-dependent manner. (**A**) This figure shows an overview of the experiments measuring the time-dependent inhibitory effect of PhoSL treatment after HBV infection. Arrows indicate the dates when conditioned media were collected for measuring HBe Ag. The blue arrows indicate the days when the medium was changed, and the red arrows indicate the days when HBV infection was performed. PhoSL was administered at the time of medium change according to the treatment period. The administration schedule for HBV+PreS1 is ① in (**A**), and for HBV is ② in (**A**). (**B**) Time course of HBeAg levels in PXB cell supernatants after HBV infection. The level of HBeAg secreted from PXB cells treated with 10 µg/mL PhoSL for various periods after HBV infection is shown as a change in absorbance. As a positive control, myristoylated PreS1 peptide was added during HBV infection. Each data point is shown with the mean +/− S.D. (**C**) The HBeAg levels secreted from PXB cells treated with 10 µg/mL PhoSL for various periods on day 12 of culture after HBV infection are shown relative to the value of the PhoSL-untreated sample on day 3 of culture, which is set at 1. (**D**) The inhibitory effect of PhoSL treatment on extracellular HBV DNA production in PXB cells infected with HBV on day 12 of culture was examined by quantitating HBV DNA recovered from the culture supernatant using real-time qPCR with HBV genomic DNA as the standard. (**E**) Intracellular HBV DNA recovered from cells on day 12 of culture was similarly quantified by real-time qPCR. (**F**) PXB cells were infected with HBV and treated with 10 µg/mL PhoSL for various periods, and cccDNA levels in intracellular HBV DNA harvested on day 12 of culture were quantified by digital PCR. Each experiment was performed with an N of 3, and the results are shown as the mean +/− standard deviation. * *p* < 0.05, ** *p* < 0.01, *** *p* < 0.005.

**Figure 3 viruses-17-01242-f003:**
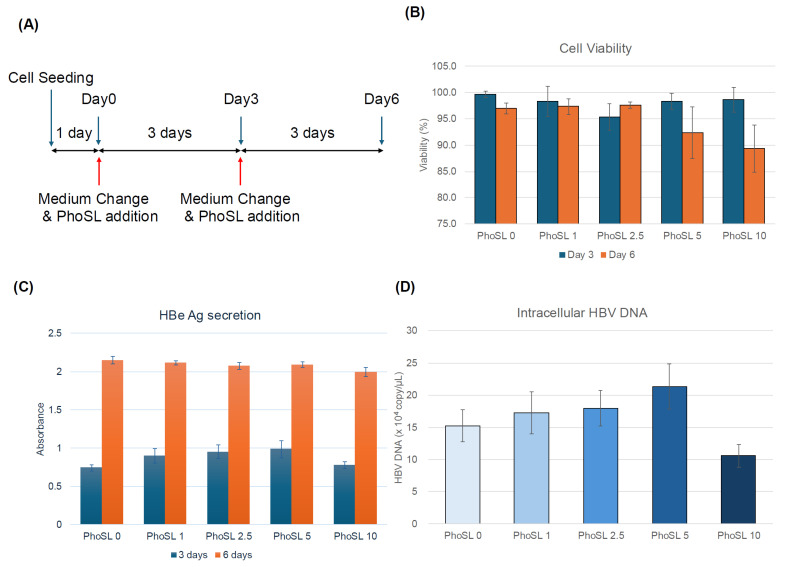
PhoSL failed to suppress the amount of HBV produced by HepG2.2.15 cells. (**A**) This figure shows an overview of the concentration-dependent experiments of the inhibitory effects of PhoSL in HepG2.2.15 cells. (**B**) PhoSL toxicity was measured by culturing cells at PhoSL concentrations of 1, 2.5, 5, and 10 µg/mL for 3 or 6 days, and staining dead cells with trypan blue. (**C**) The HBeAg levels secreted by HepG2.2.15 cells treated with 0, 1, 2.5, 5, and 10 µg/mL PhoSL for 3 or 6 days. Each data point is shown with the mean value +/− S.D. (**D**) Intracellular HBV DNA recovered from HepG2.2.15 cells on day 6 of culture was quantified by real-time PCR using HBV genomic DNA as a standard. (**E**,**F**) Extracellular HBV DNA was extracted from the supernatant of HepG2.2.15 cells harvested on day 3 and 6 of culture and quantified by real-time PCR using HBV genomic DNA as a standard. (**G**) HepG2.2.15 cells were treated with 0, 1, 2.5, or 5 µg/mL of PhoSL for 6 days, then harvested, and cccDNA levels were quantified by digital PCR using the extracted DNA. Each experiment was performed in triplicate, and the results are shown as the mean +/− standard deviation.

**Figure 4 viruses-17-01242-f004:**
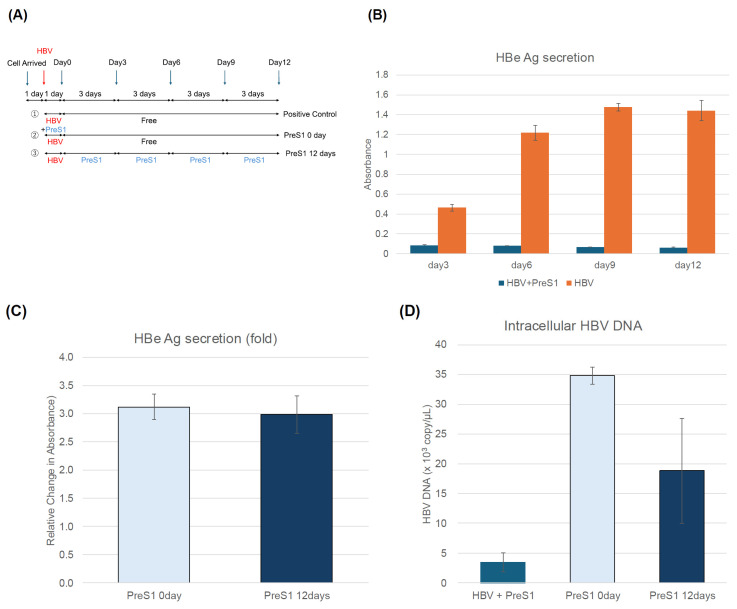
PreS1 peptide did not affect extracellular HBeAg production and extracellular HBV DNA generation, nor intracellular HBV DNA and cccDNA production in PXB cells when administered after HBV infection. (**A**) This figure shows an overview of the experiments measuring the inhibitory effect of myrPreS1 treatment after HBV infection. Arrows indicate the dates when conditioned media were collected for measuring HBe Ag. The blue arrows indicate the days when the medium was changed, and the red arrows indicate the days when HBV infection was performed. MyrPreS1 was administered continuously at medium changes for 12 days after HBV infection. The administration schedule is ① in (**A**) when PreS1 is allowed to coexist only during HBV infection, and ② in (**A**) when there is only HBV infection and PreS1 is not treated at all. (**B**) The time course of HBeAg levels in the supernatant of PXB cells with or without myrPreS1 treatment during HBV infection. (**C**) The HBe antigen levels secreted by PXB cells treated with 1 µM myrPreS1 for 0 or 12 days after HBV infection on day 12 of culture are shown as relative values, with the value of the myrPreS1-untreated sample on day 3 of culture set at 1. (**D**) Quantification of intracellular HBV DNA by real-time PCR using HBV genomic DNA as a standard. (**E**) Quantification of extracellular HBV DNA in HBV-infected PXB cells according to the duration of myrPreS1 treatment. Each data point is shown with the mean value +/− S.D. N = 3. (**F**) Quantification of cccDNA levels by digital PCR in PXB cells treated with 1 µM myrPreS1 for various periods after HBV infection. Each experiment was performed with an N of 3 and the results are shown as the mean +/− standard deviation.

## Data Availability

Dataset available on request from the authors.

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
