# Peer review of "Potential for Core Fucose-Targeted Therapy Against HBV Infection of Human Normal Hepatocytes"

_viruses, 2025, doi:10.3390/v17091242_

Round 1
Reviewer 1 Report
Comments and Suggestions for Authors
This study investigates the antiviral activity of PhoSL against HBV infection in normal human hepatocytes using PXB cells. The findings suggest that PhoSL inhibits HBeAg, HBV DNA, and cccDNA levels by binding to the HBV receptor and blocking entry through inhibition of EGFR phosphorylation. While the study presents interesting findings, several areas require clarification and revision for improved readability.
Specific comments:
Since the study analyzed the inhibitory function of PhoSL in normal human hepatocytes, it needs to include images of HBV-infected PXB cells to visually represent the inhibitory effects of PhoSL. This would complement the current bar and line graphs.
The conclusion states that PhoSL inhibits HBV entry through "inhibition of EGFR phosphorylation." However, data related to EGFR phosphorylation are not found in this study. This claim needs to be either supported with presented data or removed from the conclusions.
Revise the use of the word "some" in statements in lines 23, 312-316, and 337) regarding unknown mechanisms.
Line 23: Change "PhoSL may also inhibit HBV proliferation in cells through some mechanism" to "PhoSL may also inhibit HBV proliferation in cells through an unknown mechanism" or "PhoSL may also inhibit HBV proliferation in cells through other mechanisms that require further investigation."
Lines 312-316 and 337: Change "considering these findings together, it is speculated that the infection-suppressing effect observed when PhoSL was administered after HBV infection is not due to inhibition of reinfection, but rather due to PhoSL that was taken up into the cells by some route and inhibited somewhere in the HBV replication pathway" to "considering these findings together, it is speculated that the infection-suppressing effect observed when PhoSL was administered after HBV infection is not due to inhibition of reinfection, but rather due to PhoSL that was taken up into the cells by an as-yet-undetermined route and inhibited a step within the HBV replication pathway."
In line 24, specify the specific names of the edible mushrooms from which PhoSL is derived, rather than the generic "edible mushrooms."
The description and presentation of Figure 2 are currently confusing and require significant revisions in both the main text and the figure legend.
In Figure 2A, the condition described as "number 1" (no PhoSL treatment after HBV infection) is currently indicated as a "positive control." If it represents HBV infection alone, it should be labeled as a negative control for the PhoSL treatment's effect. Data related to scheme "number 1" is not shown in the figure. All described conditions in the figure 2A must have corresponding data presented.
In Figure 2B, the y-axis is labeled "absorbance" without further specific description. It needs to clearly indicate what is being measured such as "HBeAg Absorbance". The bars are simply indicated as "HBV." If this represents the time course of HBeAg production in HBV infection alone, this needs to be clearly explained in the figure legend and text.
In Figure 2C, the title "Relative changes on the 12th day compared to the 3rd day" above the graph is highly confusing and lacks context. Provide a clear rationale for this specific comparison in the text and figure legend. If it's a comparison between untreated and treated cells, the description needs to be rewritten for better understanding.
In Figure 2D and 2F, data for HBV infection alone (without PhoSL treatment) needs to be added for both HBV DNA and cccDNA levels to provide a proper baseline for comparison.
In Figure 3A, the "positive control" should be labeled as a negative control.
In Figure 3C, explain the rationale for including a title above only Figure 3C. Ensure consistent formatting across all figures; either add titles to all relevant figures or remove this specific one if it's redundant with the figure legend.
The authors indicated that PhoSL does not affect the secretion of hepatitis antigen particles, showing only HBeAg in Figure 3C. To fully support this claim, HBsAg levels also need to be analyzed and presented for this purpose.
Reviewer 2 Report
Comments and Suggestions for Authors
Hepatitis B virus (HBV) infection poses a significant global public health threat, contributing to considerable morbidity and mortality worldwide. Core fucose is one of the key glycans implicated in HBV infection. PhoSL, a lectin purified from edible mushrooms, has been reported to inhibit HBV infection in hepatocellular carcinoma cells by interfering with NTCP or direct HBV binding. In this study, Takamatsu et al. investigated whether PhoSL exerts a similar inhibitory effect on HBV infection in normal human hepatocytes (PXB cells). They demonstrated that PhoSL treatment, either during or after HBV infection, suppresses HBe antigen production and reduces cccDNA levels in a manner dependent on both PhoSL concentration and treatment duration. Overall, this manuscript provides valuable insights into a potential novel antiviral strategy against HBV infection. However, some claims are not fully supported by the presented data and require additional experiments to validate the conclusions.
- In Figure 1B, the legend mentions 0.5 μg/mL PhoSL, but there is no corresponding bar in the graph. Please revise accordingly.
- In the LDH-Glo cytotoxicity assay shown in Figure 1B, the error bars appear too large. To clearly demonstrate the absence of cytotoxicity, it may be advisable to repeat the experiment. Furthermore, in the Day 6 graph, although the overall error bars are smaller, the bar for PhoSL 2.5 μg/mL appears to differ significantly from the PhoSL 0 μg/mL control. Please clarify whether this difference is statistically significant.
- In Figure 1D, there appears to be a dose-dependent decrease in cccDNA levels with increasing concentrations of PhoSL. However, the current data are based on n = 1. It is recommended to increase the number of replicates to n = 3 for statistical robustness.
- The results for Figure 1D are not described in the main text. Please include a description in the Results section.
- In Figure 2D, the error bars for the "PhoSL 3 days" condition are quite large. To enable accurate interpretation of the results, it is desirable to obtain more reproducible data.
- In line 311, the authors state that "normal hepatocytes such as PXB cells hardly express the core fucosyltransferase Fut8." Please provide experimental evidence showing the low expression of Fut8 in PXB cells (e.g., mRNA or protein level comparison with other cell types), or cite a previous publication if this has already been demonstrated.
- In line 315, the authors describe that "PhoSL was taken up into the cells by some route and inhibited somewhere in the HBV replication pathway." To support this claim, please consider evaluating the effect of PhoSL on cccDNA levels in HepG2.2.15 cells, which stably replicate HBV, by treating them with PhoSL.
- In Figure 3B, although HBV alone is shown as a positive control in Figure 3A, it is unclear whether "HBV + PreS1" refers to PreS1 treatment on Day 0 or Day 12. Please clarify this in the figure or legend.
- The interpretation of the results in Figures 3C and 3D is unclear. In Figure 3A, PreS1 treatment on Day 0 is used to block infection at the time of viral entry, and this seems sufficient to prevent HBV infection as shown in Figures 2 and 3. However, in Figures 3C and 3D, HBeAg and secreted HBV DNA levels increase significantly by Day 12 compared to Day 3 under the PreS1 Day 0 condition. Which condition in Figures 3C–F corresponds to the positive control in Figure 3A? Also, please clarify the treatment protocol for "HBV + PreS1" in Figure 3D-F.
- The error bars in Figure 3E appear quite large, which may undermine the reliability of the data. Please address this issue.
- Based on Figure 3F, why is the cccDNA level higher in the PreS1 12 days condition (where reinfection is suppressed) compared to the PreS1 0 day condition? Please provide a rationale.
- In Figure 4, the early stages of infection are discussed. It would be helpful to demonstrate PhoSL binding to NTCP and/or to HBV particles in a clear and visual manner.
- To investigate the effect of PhoSL on denucleation, it would be useful to conduct experiments analyzing the levels of rcDNA in cytoplasmic and nuclear fractions. Additionally, have Southern blot analyses of rcDNA and cccDNA been performed in the context of Figures 2 and 3?
- In line 336, the authors write: “PhoSL taken up into cells by endocytosis together with HBV may also inhibit cccDNA synthesis by some mechanism.” Is the intracellular amount of PhoSL different between HBV-infected and uninfected cells? Please clarify.
Round 2
Reviewer 1 Report
Comments and Suggestions for Authors
All comments are addressed.
Author Response
The authors indicated that PhoSL does not affect the secretion of hepatitis antigen particles, showing only HBeAg in Figure 3C. To fully support this claim, HBsAg levels also need to be analyzed and presented for this purpose.
Thank you for your valuable opinion. We measured HBeAg and extracellular DNA to conclude that PhoSL inhibits cccDNA synthesis before and after HBV replication, thereby suppressing subsequent HBV replication. Because the reduction in antigen levels is a secondary effect, we believe that measuring HBeAg, which is commonly measured as a marker reflecting transcription, translation, and replication, is sufficient.